# Randomised controlled study on the effects of pilates exercises in soccer: Comparing mat and reformer methods on physical and technical performance

Osman Yılmaz[1], Turgut Kaplan[2], Ladislav Batalik[3,4,5,6]*

1 Osmaniye Korkut Ata University, School of Physical Education and Sports, Osmaniye, Turkey, 2 Retired formally at Selcuk University Faculty of Sport Science, Konya, Turkey, 3 Department of Physiotherapy and Rehabilitation, Faculty of Medicine, Masaryk University, Brno, Czech Republic, 4 Department of Rehabilitation, University Hospital Brno, Brno, Czech Republic, 5 Department of Public Health, Faculty of Medicine, Masaryk University, Brno, Czech Republic, 6 Department of Rehabilitation, Faculty of Medicine, Masaryk University, Brno, Czech Republic

* batalik.ladislav@fnbrno.cz

## Abstract

This study was conducted to determine and compare the effects of reformer pilates (RP) and mat pilates (MP) exercises on soccer players' physical parameters and technical skills. Thirty voluntary participants were randomly assigned to either RP group (n = 10; age = 20.60 ± 1.65), MP group (n = 10; age = 19.40 ± 1.35) and control group (CG) (n = 10; age = 20.10 ± 1.15). Technical and physical performance tests were performed. In the RP group counter movement jump (CMJ), standing broad jump (SBJ), single leg triple hop right-left, balance right-left leg, flexibility, 10-20m sprint, german agility (GA), speed dribbling (SPD), loughborough soccer passing (LSPT), lobbed passing right food, lobbed passing total measurement results showed a statistically significant difference between pre-test and post-test mean values (p < 0.05). In the MP group, balance right-left leg, single leg triple hop right-left, 5 - 10m sprint, GA, SPD, LBP, lobbed passing right measurement results showed a statistically significant difference between pre-test and post-test mean values (p < 0.05). The control group had no significant difference in the pre-test and post-test mean values of technical and physical performance measurements (p > 0.05). Between-group comparisons revealed superior improvements in GA, LSPT, and single-leg triple hop right-left in the RP group compared to the MP group. Based on these findings, coaches and sports performance specialists may enhance athletes' physical performance and technical skills by incorporating Pilates exercises (particularly RP) into training programs.

**Data availability statement:** All relevant data can be found at the following Zenodo page: Yılmaz, O., & Kaplan, T. (2023). Randomised Controlled Study on the Effects of Pilates Exercises in Soccer: Comparing Mat and Reformer Methods on Physical and Technical Performance. https://doi.org/10.5281/zenodo.15281851.

**Funding:** This research was funded by the Ministry of Health, Czech Republic; conceptual development of research organization (FNBr, 65269705). The funders had no role in study design, data collection and analysis, decision to publish, or preparation of the manuscript.

**Competing interests:** The authors have declared that no competing interests exist.

## Introduction

The primary objective for coaches, sports physiotherapists, and players is to enhance athletic performance to achieve competitive success [1]. Athletic performance requirements differ significantly based on the specific sports disciplines [2]. In soccer, optimal performance demands multifaceted physical and technical abilities, including repeated high-intensity sprints, endurance, agility, coordination, dynamic balance, and precise execution of technical skills such as dribbling, passing, shooting, and heading [3–5] Soccer players must possess a high degree of balance, coordination, flexibility, agility, speed, endurance, and technical proficiency to perform effectively [6]. Therefore, training interventions targeting these specific components are critical to improving overall soccer performance.

Pilates exercises represent a promising training methodology due to their unique emphasis on core strength, muscular integration, flexibility, and neuromuscular coordination, aspects directly relevant to soccer players' performance [7,8] Pilates is designed to create a strong, harmonious, flexible, and balanced body by integrating major and minor muscle groups through precise and controlled movements [9]. Pilates training is typically executed through two primary methods: mat Pilates, which uses body weight and supplementary materials (e.g., small balls, Pilates rings), and apparatus-based Pilates, notably using a reformer machine. The reformer utilizes springs for controlled force and resistance, enabling diverse exercises in lying, sitting, or standing positions, thereby enhancing external resistance during muscular contractions [10–12].

Recent literature has demonstrated Pilates' effectiveness in improving performance-related parameters across various sports, including static and dynamic balance in archers [13], muscular endurance and technical skills in volleyball players [14], posture and flexibility in dancers [15], coordination and agility in badminton athletes [16], and core strength in cricket and karate athletes [17,18]. Despite these findings, evidence specifically examining Pilates' efficacy in soccer is limited. Previous studies on soccer have primarily assessed singular Pilates modalities without directly comparing different Pilates methods or comprehensively evaluating both physical and technical performance outcomes within one design [19–21].

To address these gaps, the present study uniquely compares two distinct Pilates modalities (mat Pilates vs. reformer Pilates) in a controlled experimental design involving amateur male soccer players. The choice of Pilates training over other modalities, such as aerobic gymnastics or traditional stretching, is justified by Pilates' superior focus on controlled dynamic movements, precise body control, integrated muscular engagement, and enhancement of core stability, factors crucial for soccer-specific actions.

We hypothesized that both mat and reformer Pilates training would significantly improve soccer players' physical performance and technical skills compared to a control group. Moreover, we expected reformer Pilates, due to its increased mechanical resistance and functional variability, to provide superior improvements in performance metrics. Thus, the current research aimed to investigate and compare the effects of reformer Pilates and mat Pilates exercises on the physical performance and technical skills of amateur soccer players.

## Materials and methods

### Participants

Thirty amateur male soccer players voluntarily participated in this study as reformer pilates group (n = 10; age = 20.60 ± 1.65; weight (kg) = 71.53 ± 5.80; height (cm) = 179.80 ± 7.58; BMI (kg/m2) = 22.12 ± 1,17), mat pilates group (n = 10; age = 19.40 ± 1.35; weight (kg) = 71.21 ± 5.28; height (cm) = 179.70 ± 4.83; BMI (kg/m2) = 22.05 ± 1.47), and control group (n = 10; age = 20.10 ± 1.15; weight (kg) = 73.15 ± 8.85; height (cm) = 179.85 ± 4.56; BMI (kg/m2) = 22.61 ± 2.55). Participants are amateur soccer players competing in local amateur league. Players were accustomed to a training regimen of at least six training units per week and had participated in soccer training and competitive matches for a minimum of 4 years. The experimental groups included players on the same team. The control group consisted players from a diffrent team. The research was executed out at the beginning of the season, during the preparation period. The participants were verbally informed of the study's content, methodologies, procedures, benefits, potential risks and permission forms were collected voluntarily. The players included in the study did not have any injuries or general health problems before the study. The players did not experience any injuries during the study. The groups consistently participated in the training sessions designed for them.

Participants were allocated to three research groups using a random assignment method: Reformer pilates (n = 10), mat pilates (n = 10), and control (n = 10) (www.randomization.com). The flow diagram is shown in Fig 1. The randomization process was conducted by an individual who was both unrelated to the objectives of the investigation and not involved in any aspect of data collecting or any phase of the trial. The study was carried out between 20.11.2021 and 10.03.2022. The study was prepared in accordance with CONSORT guidelines. To determine the appropriate sample size for this study comparing the effects of RP and MP on physical performance and technical skills in amateur male soccer players, a power analysis was conducted. The study aimed to detect a significant difference in physical performance and technical parameters, which are continuous variables. Based on previous literature and pilot data, an effect size of 0.8 was estimated for physical performance and technical skills improvements between the intervention groups, indicating a large effect. To achieve a power of 0.80 and a significance level (alpha) of 0.05, it was calculated that a minimum of 10 participants per group would be required, totaling 30 participants across the three groups (RP, MP, and control). This sample size allows for sufficient power to detect significant differences in physical performance and technical skills outcomes while accounting for potential dropouts. Given the experimental design and the exploratory nature of comparing two different Pilates methods, this sample size is considered adequate for providing reliable and generalizable results within the context of this study. This study was conducted in accordance with the principles of the Declaration of Helsinki and approved by the ''Selcuk University" Ethics Committee (11.11.2021 - E174010)

### Study desing

The present study employed a three-group (reformer pilates, mat pilates and control group), matched, experimental design. The research was conducted in three phases. The study was completed over a total of 10 weeks, consisting of 1 weeks of pre-testing, 8 weeks of pilates training interventions (Reformer pilates and Mat pilates), and 1 weeks of post-testing. The players completed a 5–10-20m sprint test, german agility test, counter movement jump (CMJ) test, standing broad jump (SBJ) test, single-leg triple hop test, sit and reach test, Y balance under extremity (YBT) test, speed dribbling test, loughborough soccer passing test and lobbed passing test before and after the 8-week intervention period. All tests and training periods were performed on an artificial grass pitch at a consistent time of day (17:00), following the same sequence of tests and players. The reformer and mat pilates group were subjected to six weekly training days for eight weeks, including three Pilates and three team training days. The control group continued their regular team training regimen six days per week. Three groups were allocated one day of rest per week. The mat and reformer Pilates group exercise sessions lasted 50–60 minutes. A 5-minute warm-up session was conducted just before starting the Pilates exercises. MP groups had two-minute rest breaks after the toe-top, spine twist, and circle side-kick movements. RP groups had two-minute rest intervals after the supine arm

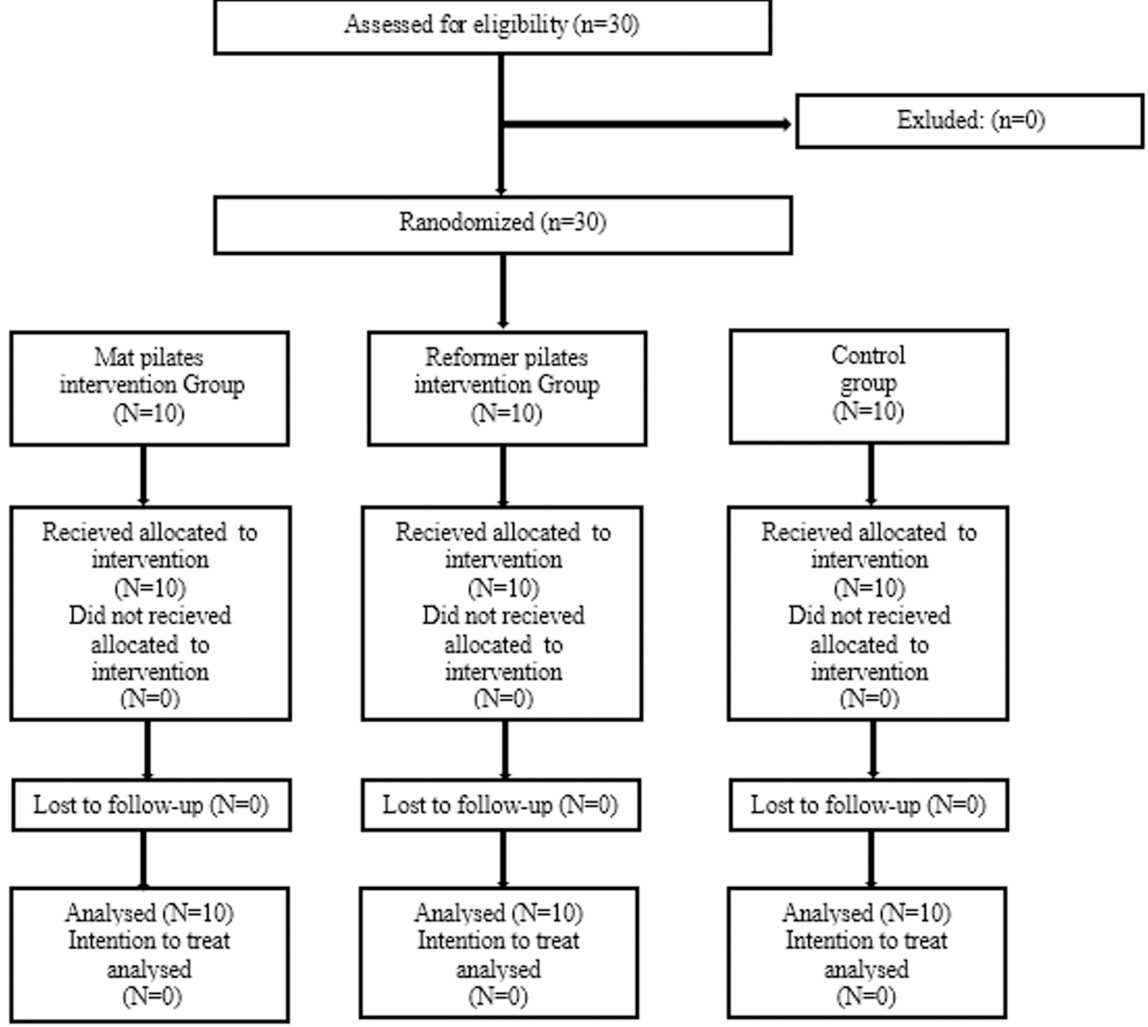

**Fig 1. Consort Flowchart Diagram.**

series, short box, arm series, and knee stretch movements. The details of the exercise schedules are presented in Table 1 and Table 2. Exercise programs prepared by an expert instructors were applied to reformer pilates and mat pilates groups. Soccer training sessions were conducted by a team coach with a UEFA B licence and lasted between 70–90 minutes.

## Physical and technical performance tests

**Sprint tests.** A standart warm-up has been completed. The participants proceeded to run a distance of 20 meters, while their running durations were measured using a set of paired photocells (Microgate, Bolzano, Italy) placed at intervals of 5, 10, and 20 meters. Participants assume an upright position with their anterior limbs positioned 0.2 meters from the initiation photocell beam. The evaluation process is completed within a matter of seconds [22].

**Counter Movement Jump (CMJ).** The participant was directed to do the counter movement jump (CMJ) with an arm swing. This involved squatting to a knee flexion angle of around 90° and executing a leap with maximal concentric contraction to achieve maximum height. The participants performed two initial leaps, succeeded by five maximum jumps, with 6-second intervals in between [23].

**Table 1. Mat pilates exercise program.**

| Mat Pilates Exercise | | | | Number of Repetitions | |
|---|---|---|---|---|---|
| | | | | 1,2,3,4 week | 5,6,7,8 week |
| 1 | ½ Roll Up | 12 | Up-Down Side Kick | 10 | 12 |
| 2 | Roll Up | 13 | Front-Back Side Kick | | |
| 3 | One Leg Circle | 14 | Circle Side Kick | | |
| 4 | Double Leg Straight Lower | 15 | Flight | | |
| 5 | One Leg Stretch | 16 | Swan | | |
| 6 | Double Leg Stretch | 17 | Rest Position | | |
| 7 | Criss Cross | 18 | Swimming | | |
| 8 | Toe Top | 19 | Push Up | | |
| 9 | Shoulder Bridge | 20 | Long Stretch | | |
| 10 | Seated Tracking | 21 | Cat Cow | | |
| 11 | Spine Twist | 22 | Mermaid Stretch | | |

**Table 2. Reformer pilates exercise program.**

| Reformer Pilates Exercise | | | | Number of Repetitions | |
|---|---|---|---|---|---|
| | | | | 1,2,3,4 week | 5,6,7,8 week |
| 1 | Footwork Series, Toes | 15 | Short Box And Arm Series, Biceps Curl | 10 | 12 |
| 2 | Footwork Series, Heels | 16 | Short Box And Arm Series, Rowing | | |
| 3 | Footwork Series, Tendon Strech | 17 | Short Box And Arm Series, Triceps | | |
| 4 | Footwork Series, V Position | 18 | Long Box Series, Swan | | |
| 5 | Supine Arm Series, Pull | 19 | Long Box Series, Pulling Fly | | |
| 6 | Supine Arm Series, Circle | 20 | Stomach Massage, Round | | |
| 7 | Supine Arm Series, Pull Head Up | 21 | Stomach Massage, Twist | | |
| 8 | Supine Arm Series, Triceps Press | 22 | Knee Stretch, Round | | |
| 9 | Short Box Series, Round | 23 | Knee Stretch, Flat Back | | |
| 10 | Short Box Series, Flat Back | 24 | Hip Work Series, Double Leg Press | | |
| 11 | Short Box Series, Twist | 25 | Hip Work Series, Hamstring Pulls | | |
| 12 | Short Box And Arm Series, Chest Fly | 26 | Hip Work Series, Leg Circle | | |
| 13 | Short Box And Arm Series, The Gift | 27 | Hip Work Series, Frog | | |
| 14 | Short Box And Arm Series,Rhomboid | 28 | Side Strech | | |

**Standing Broad Jump (SBJ).** The standing broad jump test was conducted on a rigid surface. Participants position themselves by aligning their heels with the starting line and ensuring that their feet are parallel to each other. Following the researcher's instructions, students execute the maneuver by leaping with maximum horizontal displacement. The tests are conducted twice, with a five-minute interval between each session. The distance was quantified using a tape measure in centimeters, from the beginning line to the heel of the foot that was closest to the starting line. The superior result of the two tries was documented [24].

**Single-leg triple hop test.** Participants start the activity by assuming a standing position on the specified foot, with hands placed on the hips and toes positioned behind the starting line. Participants are directed to execute a maximum of 3 forward leaps, initially landing on the same leg, with the aim of reducing the amount of time their feet spend on the ground following the first and second jumps. When completing the last jump, participants are instructed to execute a controlled fall

and maintain this position for a duration of 2 seconds. Failure to land invalidates the attempt and the leap is retaken after a 60-second rest [25].

**Sit and reach test (SRT).** The measurement of flexibility using the traditional SRT methodology. In this experiment, a conventional SRT box of 30.5 cm in height was employed. The box was equipped with a sliding reach indication on a measuring scale ranging from 0 to 50 cm. The 35 centimeter mark is in perfect alignment with the foot panel of the box. The subject performs testing in a seated position on the floor, with their legs together, knees extended, and bare soles pressed on the foot panel of the testing box. The participant is instructed to maintain an open position of their knees during the test and to ensure the stability of their knees during the test. The maximal position should be progressively achieved and sustained for a duration of 2 seconds. A total of two measures were recorded for each player, with a time interval of 30 seconds between each measurement [26].

**Y balance under extremity (YBT).** The YBT test kit is organized in three directions for the lower extremities: anterior, posteromedial, and posterolateral [27]. To score each of the three aspects, they are normalized by the participant's leg length. The composite score is determined by calculating the mean of the right and left reach distances for each of the three directions, and then adding together the averages for all three reach directions. The score is calculated by dividing it by three times the average length of the legs and arms, and then multiplying it by 100 [27].

**German agility.** This agility test is one of the tests used to determine motor diagnostics in the German football talent identification and development program. The participant begins the race from a stationary position, without any cues, and navigates through a slalom course to reach the finish line as quickly as possible. It is utilized on a track measuring 13 meters in length and 4 meters in wide. Slalom bars are positioned at the 3rd, 4th, and 5th meters along the 4th wide meter and a long with at the 0th, 8th, 9th, and 10th meters. The slalom bars at the 4th and 9th meters are positioned 0.50 cm inward You can access the test visual in the validity and reliability article [28].

## Speed dribbling test

This test measures coordinated speed dribbling and timed speed. The athlete starts behind the line with the ball at "Ready-Go". From 5 m, he dribbles right around the first triangle post (2). He dribbles around the other posts in sequence. He dribbles around a block after 10 m. After 8 m, he plays the ball around one side of a square (4), steps on it, rushes from the other side (5), takes it (6), and races toward the photocell door. Expert estimates time from "Go" till player puts ball under feet. Stopwatches measure 0.1 seconds [29].

**Lobbed passing test.** Pass the ball to any area 30 meters from where the player is standing. The ball may roll or be motionless when passed. In the middle of the smallest square, another player gets the ball. Another player scores when they receive an aerial pass in any of the three squares. The ball lands in the smallest inner square 3 points; in the 3m x 3m square, 2 points; outer square; 1 point. Each leg gets 5 tries. Ball landing squares determine points [30].

**Loughborough soccer passing test.** Participants make 16 passes, 4 per color, during the test. Pass accuracy, timing, and penalties were recorded. Athletes must finish the test quickly and accurately. Eight passes are performed to the long (green-blue) and short (white-red) targets in each trial [31]. Test error duration; +5 seconds are added to the time for a pass that does not hit the target stand or is thrown to the wrong target. If a pass fails to reach the target center, +3 seconds will be added total duration. When the ball is manually touched, +3 seconds are added to the total duration. When the passes the ball to the target from outside the passing area, +2 seconds are added to the duration. When you touch each cone, +2 seconds are added to the duration. Every second that the test time (43 seconds) was exceeded was added to the total passing time. For passes finding the target, -1 second was subtracted from the total time [31].

**Statistical analysis.** Data were presented as the mean ± standard deviation (SD). The assumptions of normality, skewness, and kurtosis were examined before using parametric tests. The distribution was within the range of -1.5 to +1.5, indicating that the data were normally distributed [32]. An independent samples t-test was used to evaluate within-group differences. Repeated-measures multivariate analysis of variance (3x2) (two-way ANOVA) was used to analyze

the data. The factors included three Pilates groups (reformer, mat, and control) and repeated measurements (pre- and post-exercise). The effect sizes for all tests in the two-way ANOVA were calculated using a partial eta squared. This is a measurement of the proportion of the variation in the dependent variable that can be attributed to the independent variables. Values of 0.01, 0.06, and 0.15 were considered small, medium, and large, respectively [33]. The significance threshold for pairwise comparisons was determined using Bonferroni's post hoc test. This is a test that is employed to ascertain whether there are notable disparities in the averages of two or more groups. Cohen's effect size (d) was used to determine the effect size in pairwise comparisons. Effect size (d) was classified as negligible (0.00–0.20), small (0.21–0.59), medium (0.60–1.19), large (1.20–1.99), very large (>2.00), and nearly perfect (>4.00) [34]. Statistical analyses were performed using the SPSS statistical software package (SPSS Inc., Chicago, IL, USA, version 26.0). Statistical significance was set at $p < 0.05$.

## Results

In the findings section of this study, the statistical results of the physical and technical performance effects of different Pilates exercises in the groups are provided. Table 3 presents the descriptive statistical information relevant to the study.

Fig 2 presents the statistical results of reformer pilates (RP), mat pilates (MP) and control within and between groups for counter movement jump (CMJ), standing broad jump (SBJ), single leg triple hop and balance. The RP group had a statistically significant difference in CMJ (p=0.001; d=-0.72 [Medium effect]), SBJ (p=0.002; d=-1.03 [Medium effect]), single leg triple hop right (p=0.001; d=-1.08 [Medium effect]), single leg triple hop left (p=0.003; d=-0.80 [Medium effect]) and balance right leg (p=0.001; d=-1.53 [Large effect]) and balance left leg (p=0.001; d=-1.16 [Medium effect]) responses. The MP group had a statistically significant difference in single leg triple hop right (p=0.002; d=-0.68 [Medium effect]), single leg triple hop left (p=0.040; d=-0.51 [Small effect]) and balance right leg (p=0.010; d=-0.99 [Medium

**Table 3. Descriptive statistical information.**

| Variables | Reformer Pilates Group | | Mat Pilates Group | | Control Group | |
|---|---|---|---|---|---|---|
| | Pre-Test X±SD | Post-Test X±SD | Pre-Test X±SD | Post-Test X±SD | Pre-Test X±SD | Post-Test X±SD |
| Flexibility | 33.45±6.35 | 35.95±6.08 | 30.20±6.64 | 31.95±6.04 | 29.55±6.03 | 30.05±6.04 |
| CMJ | 41.22±5.59 | 45.33±5.77 | 38.38±6.27 | 40.72±8.07 | 37.22 ±5.97 | 37.52±5.90 |
| SBJ | 212.80±12.07 | 225.60±12.76 | 207.90±21.93 | 207.10±24.88 | 207.80±20.39 | 207.40±19.72 |
| SLTHT R | 580.60±34.88 | 620.10±37.93 | 500.70±65.00 | 544.70±63.72 | 507.30±74.43 | 516.20±50.46 |
| SLTHT L | 570.20±41.64 | 603.70±42.43 | 507.00±98.30 | 551.50±76.40 | 498.80±90.21 | 506.40±77.13 |
| 5m sprint | 0.99±0.07 | 0.95±0.07 | 1.06±0.73 | 0.99±0.91 | 1.05±0.09 | 1.03±0.08 |
| 10m sprint | 1.77±0.08 | 1.67±0.06 | 1.78±0.11 | 1.69±0.09 | 1.79±0.12 | 1.79±0.10 |
| 20m sprint | 3.08±0.10 | 2.99±0.09 | 3.02±0.17 | 3.00±0.15 | 3.09±0.17 | 3.08±0.18 |
| German Agility | 7.55±0.24 | 7.40±0.27 | 7.91±0.34 | 7.74±0.29 | 7.86±0.30 | 7.85±0.32 |
| Balance Right Leg | 82.53±8.15 | 93.18±5.50 | 79.64±10.06 | 89.14±9.20 | 77.65±10.81 | 78.21±10.75 |
| Balance Left Leg | 83.32±10.04 | 93.30±6.84 | 80.67±11.90 | 89.96±7.66 | 79.28±9.65 | 80.22±9.24 |
| Speed Dribling | 23.47±2.49 | 21.67±0.73 | 24.05±1.37 | 22.58±0.93 | 22.05±1.37 | 22.17±1.34 |
| LSPT | 51.55±10.71 | 39.71±9.09 | 58.64±12.29 | 45.90±9.52 | 56.79±12.17 | 57.39±11.49 |
| Lobbed Passing R | 3.90±1.97 | 6.60±3.27 | 3.50±2.46 | 5.00±2.54 | 3.80±1.48 | 4.10±1.10 |
| Lobbed Passing L | 2.40±1.96 | 2.70±1.95 | 2.10±2.47 | 2.20±2.90 | 2.20±1.62 | 1.10±2.33 |
| Lobbed Passing Total | 6.30±3.40 | 9.30±3.77 | 5.60±4.06 | 7.20±2.39 | 6.00±2.67 | 5.20±2.04 |

LSPT: Loughborough Soccer Passing Test, CMJ: Counter Movement Jump, SBJ: Standing Broad Jump, SLTHT: Single Leg Triple Hop Test R: Right, L: Left, * Significant difference between groups post-training, X: mean, SD: Standart deviation

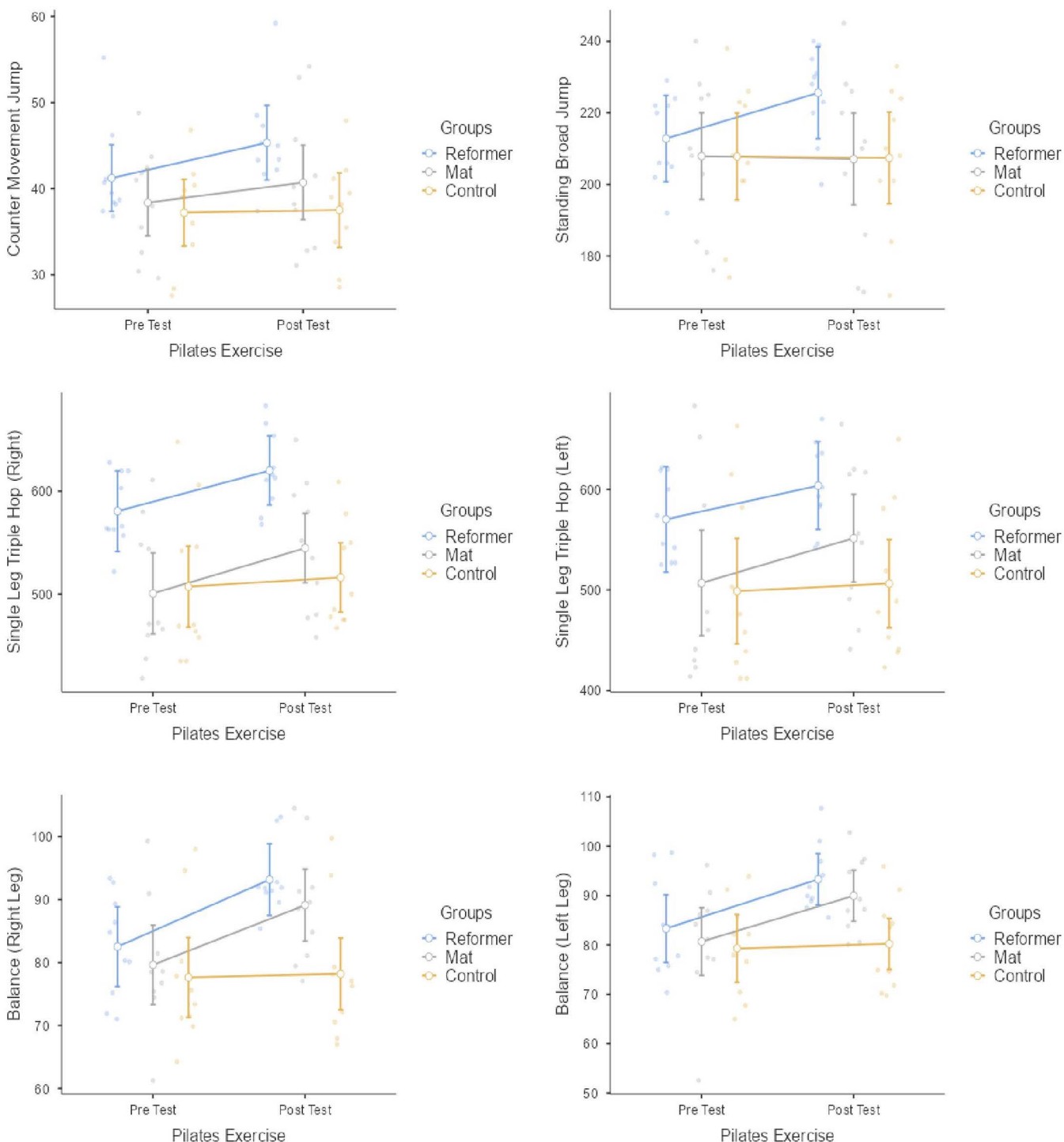

**Fig 2. Counter movement jump, standing broad jump, single leg triple hop and balance responses to pilates exercises.**

effect]) and balance left leg (p = 0.021; *d* = -0.93 [Medium effect]) responses. No statistically significant differences were found in CMJ and SBJ in the MP group (p > 0.05). No significant difference were observed in the pre-post test values of the control group (p > 0.05). Significant differences were observed between the groups in the single leg triple hop right (F = 7.875, p < 0.002, np2 = 0.368) and single leg triple hop left (F = 3.612, p < 0.041, np2 = 0.211) outcomes. Post-hoc tests showed that RP exercise better improve in single leg triple hop right and left performance compared to MP exercise There were no significant differences we observed between the groups in CMJ, SBJ, balance right leg and balance left leg results (p > 0.05) (Fig 2).

Fig 3 presents the statistical results of RP group, MP group and control group within and between groups for flexibility, agility and sprint performance response. The RP group had a statistically significant difference in flexibility (p = 0.001; *d* = -0.40 [Small effect]), agility (p = 0.010; *d* = 0.59 [Small effect]) and 10m sprint (p = 0.002; *d* = 1.41 [Large effect]) and 20m sprint (p = 0.005; *d* = 0.95 [Medium effect]) responses. No statistically significant differences were found in 5m sprint in the RP group (p > 0.05). The MP group had a statistically significant difference agility (p = 0.006; *d* = 0.54 [Small effect]) 5m sprint (p = 0.008; *d* = 0.08 [Negligible effect]) and 10m sprint (p = 0.006; *d* = 0.90 [Medium effect]). No statistically significant differences were found in flexibility and 20m sprint in the MP group (p > 0.05). No significant difference were observed in the pre-post test values of the control group (p > 0.05). Significant differences were observed between the groups in the agility (F = 5.184, p < 0.012, np2 = 0.277). Post-hoc tests showed that RP exercise better improve in agility performance compared to MP exercise. There were no significant differences we observed between the groups in flexibility, 5m sprint, 10m sprint and 20m sprint performance results (p > 0.05) (Fig 3).

Fig 4 presents the statistical results of RP group, MP group and control group within and between groups for speed dribbling, loughbrough soccer passing, lobbed passing right/left leg and lobbed passing total point. The RP group had a statistically significant difference in speed dribbling (p = 0.040; *d* = 0.98 [Medium effect]), loughbrough soccer passing (p = 0.009; *d* = 1.19 [Medium effect]), lobbed pass right leg (p = 0.008; *d* = 1.00 [Medium effect]) and lobbed passing total point (p = 0.011; *d* = 0.84 [Medium effect]). No statistically significant differences were found in lobbed passing left legin the RP group (p > 0.05). The MP group had a statistically significant difference in speed dribbling (p = 0.001; *d* = 1.26 [Large effect]), loughbrough soccer passing (p = 0.014; *d* = 1.16 [Medium effect]) and lobbed passing right leg (p = 0.048; *d* = 0.60 [Medium effect]). No statistically significant differences were found in lobbed passing left leg and lobbed passing total point in the MP group (p > 0.05). No significant difference were observed in the pre-post test values of the control group (p > 0.05). Significant differences were observed between the groups in the loughbrough soccer passing (F = 3.504, p < 0.044, np2 = 0.206). Post-hoc tests showed that RP exercise better enhancement in loughbrough soccer passing performance compared to MP exercise. There were no significant differences we observed between the groups speed dribbling, lobbed passing right/left leg and lobbed passing total point performance results (p > 0.05).

## Discussion

This study aimed to determine and compare the effects of RP and MP exercises on physical and technical performance in amateur soccer players. RP within-group evaluation showed significant improvements in flexibility, counter movement jump (CMJ), standing broad jump (SBJ), single-leg triple hop test (right-left), 10m and 20m sprints, german agility, balance (right-left leg), speed dribbling, loughborough soccer passing, lobbed passing (right foot), and total lobbed pass points. No significant improvements were observed in the 5m sprint and lobbed passing (left foot). Although previous literature on RP exercises in soccer is scarce, studies in other athletic populations report similar enhancements in CMJ, agility, and balance [10,35–37]. These improvements may be attributed to RP exercises' focus on core strength, flexibility, and proprioceptive ability, critical for explosive movements and precise technical executions.

Significant improvements in technical skills, such as speed dribbling and passing accuracy, observed in the RP group may be associated with enhanced lower extremity balance, explosive power, agility, and speed. Supporting evidence suggests a strong relationship between these physical attributes and soccer-specific technical performance [38,39].

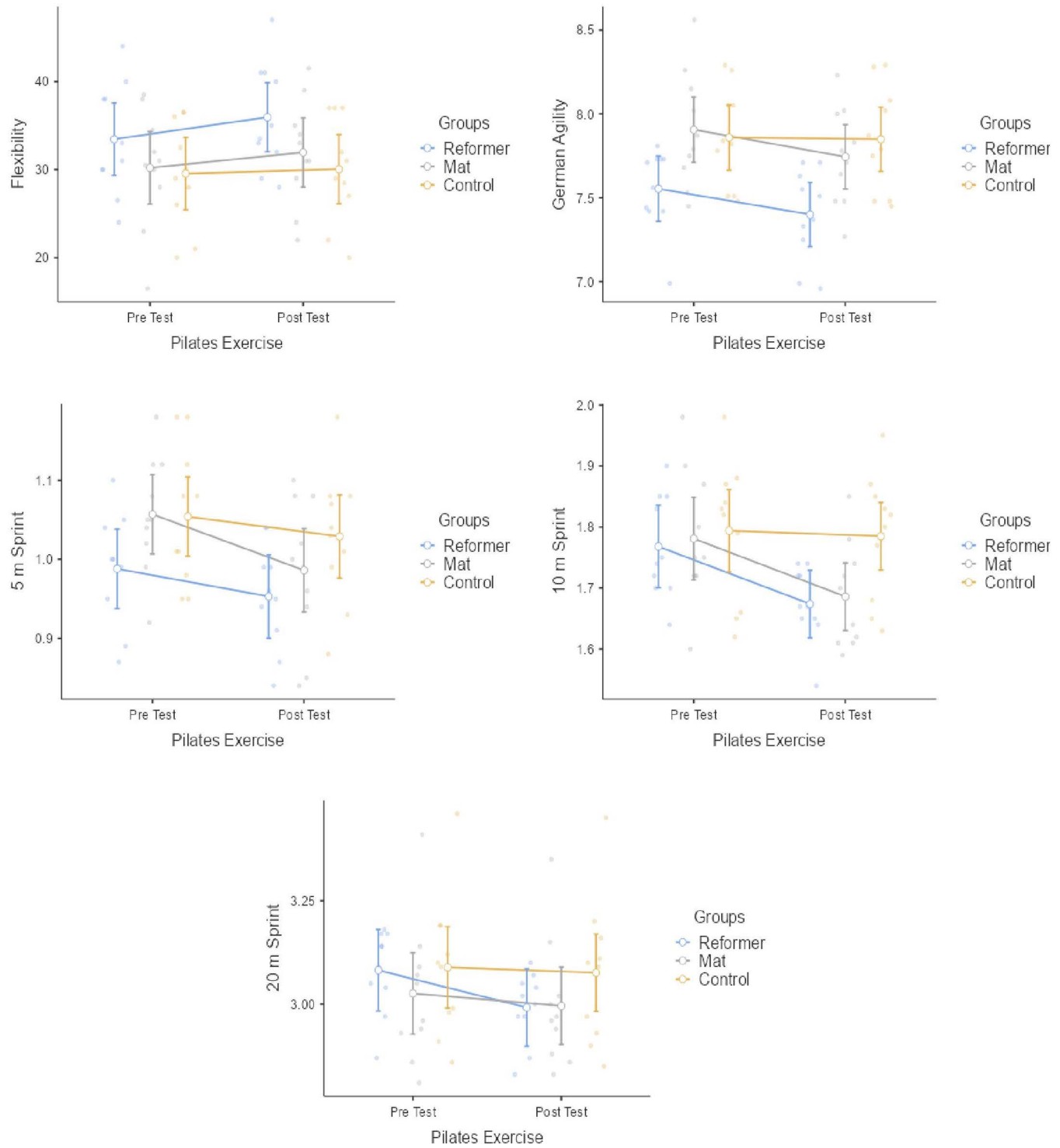

**Fig 3. Flexibility, agility and sprint responses to pilates exercises.**

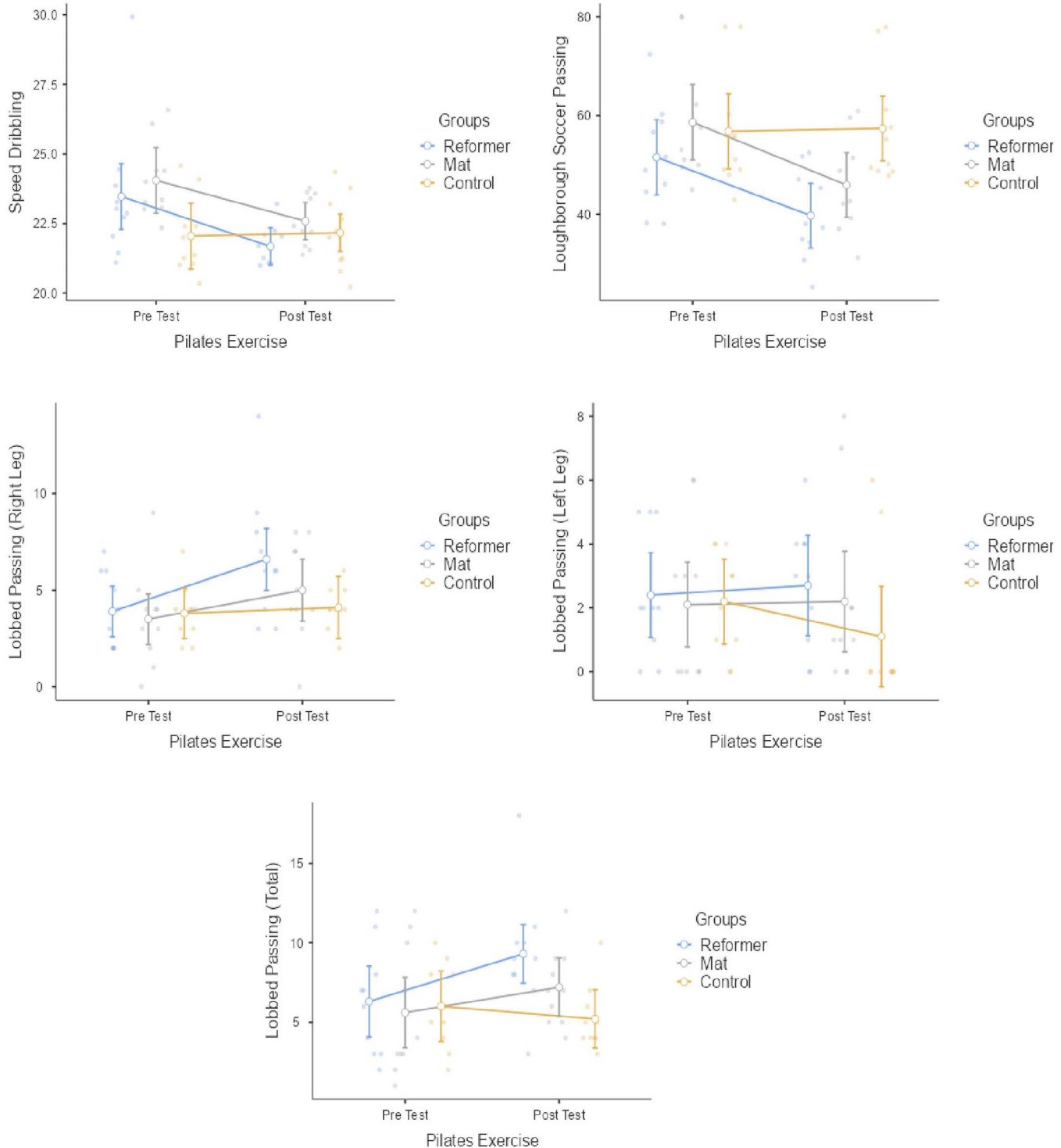

**Fig 4. Tecnichal responses to pilates exercises.**

The MP within-group evaluation revealed improvements in single-leg triple hop (right-left), agility, balance (right-left leg), speed dribbling, loughborough soccer passing, lobbed passing (right foot), 5m and 10m sprints. However, MP did not significantly affect flexibility, CMJ, SBJ, 20m sprint, lobbed passing (left foot), or total lobbed passing points. Similar studies confirm the beneficial effects of MP exercises on agility, speed, balance, and technical skills across various sports [16,19,40,41] These findings highlight the potential of MP exercises to improve athletes' balance, proprioception, and controlled movements, positively influencing soccer-specific skills and short-distance speed.

The between-group comparison demonstrated RP exercises were more effective than MP exercises in improving single-leg triple hop (right-left), german agility, loughborough passing accuracy. No significant between-group differences were noted in other tested parameters. The superiority of RP exercises may result from the eccentric-concentric contraction mechanisms involved, higher neuromuscular activation, and increased mechanical resistance provided by the reformer, which enhances explosive power, agility, and technical precision [42,43].

## Practical implications

This study demonstrated that both RP and MP exercises effectively enhance physical and technical performance in amateur soccer players, with RP offering greater benefits in specific parameters such as agility, explosive power, and passing accuracy. Coaches and practitioners can apply these findings by selecting Pilates training methods aligned with their athletes' specific performance goals. MP exercises offer accessibility and practicality for implementation in limited spaces, whereas RP provides greater exercise diversity and resistance, suitable for targeted performance enhancement.

## Limitations and future research

Several limitations of this study should be acknowledged. The sample consisted exclusively of young male amateur soccer players, limiting generalizability. Future research should include diverse participant groups regarding age, gender, competitive levels, and playing positions. Additionally, the eight-week training duration and frequency (three days per week) may not reflect the full potential of Pilates interventions. Further research should explore longer intervention durations and varying training frequencies. The lack of nutrition control also represents a limitation. Future studies incorporating nutritional monitoring could further elucidate Pilates' effects on soccer performance. Addressing these factors in future studies would strengthen the evidence base for Pilates training in soccer.

## Conclusions

This study provides evidence supporting the implementation of Pilates exercises, especially reformer Pilates, as effective training modalities to enhance physical and technical performance among amateur soccer players. Both Pilates modalities improved performance outcomes; however, reformer Pilates exhibited superior effects due to its greater mechanical resistance and exercise variety. Coaches should consider reformer Pilates when targeting significant enhancements in agility, explosive power, and technical skills. Future research is warranted to further refine training protocols and expand the applicability of these findings to broader athlete populations.

## Author contributions

**Conceptualization:** Osman Yılmaz, Turgut Kaplan.

**Data curation:** Osman Yılmaz.

**Formal analysis:** Osman Yılmaz.

**Funding acquisition:** Osman Yılmaz.

**Investigation:** Osman Yılmaz.

**Methodology:** Osman Yılmaz.

Project administration: Osman Yılmaz.

Resources: Osman Yılmaz.

Software: Osman Yılmaz.

Supervision: Osman Yılmaz.

Validation: Osman Yılmaz.

Visualization: Osman Yılmaz, Turgut Kaplan, Ladislav Baťalík.

Writing – original draft: Osman Yılmaz.

Writing – review & editing: Osman Yılmaz, Turgut Kaplan, Ladislav Baťalík.

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
