## [Decision Letter · Decision Letter 0]

9 Mar 2025

PONE-D-25-04295Randomised Controlled Study on the Effects of Pilates Exercises in Soccer: Comparing Mat and Reformer Methods on Physical and Technical PerformancePLOS ONE

Dear Dr. Yılmaz,

Thank you for submitting your manuscript to PLOS ONE. After careful consideration, we feel that it has merit but does not fully meet PLOS ONE’s publication criteria as it currently stands. Therefore, we invite you to submit a revised version of the manuscript that addresses the points raised during the review process.

We look forward to receiving your revised manuscript.

Kind regards,

Julio Alejandro Henriques Castro da Costa

Academic Editor

PLOS ONE

Journal Requirements:

2. Peer review at PLOS GPH is not double-blinded (https://journals.plos.org/globalpublichealth/s/editorial-and-peer-review-process). For this reason, authors should include in the revised manuscript all the information removed for blind review.

For additional information about PLOS ONE ethical requirements for human subjects research, please refer to http://journals.plos.org/plosone/s/submission-guidelines#loc-human-subjects-research .

6. In the online submission form, you indicated that data can be obtained for research purposes by making a reasonable request to the corresponding author.

7. Please amend your list of authors on the manuscript to ensure that each author is linked to an affiliation. Authors’ affiliations should reflect the institution where the work was done (if authors moved subsequently, you can also list the new affiliation stating “current affiliation:….” as necessary).

8. Please ensure that you refer to Figure 1 in your text as, if accepted, production will need this reference to link the reader to the figure.

Reviewers' comments:

Reviewer's Responses to Questions

**Comments to the Author**

1. Is the manuscript technically sound, and do the data support the conclusions?

Reviewer #1: No

Reviewer #2: Yes

Reviewer #3: Yes

2. Has the statistical analysis been performed appropriately and rigorously? 

Reviewer #1: No

Reviewer #2: Yes

Reviewer #3: Yes

3. Have the authors made all data underlying the findings in their manuscript fully available?

Reviewer #1: Yes

Reviewer #2: No

Reviewer #3: Yes

4. Is the manuscript presented in an intelligible fashion and written in standard English?

Reviewer #1: Yes

Reviewer #2: Yes

Reviewer #3: Yes

5. Review Comments to the Author

Reviewer #1: Dear authors:

The following are a number of issues that prevent the manuscript from being suitable for publication. I hope you will understand that all of them are made with the sole purpose of helping you to improve your work.

Introduction

- There is a lack of contextualisation of football in the work. It is not possible to present a work on improving the performance of a sport without going into the aspects of performance in depth.

- The bibliographic selection is scarce and outdated and does not provide relevant information to contextualise the study. There are a multitude of academic studies of great impact on performance in football that are not present in the article.

- The authors point out that there is no evidence of performance improvement in football players through Pilates and focus on the differences between two Pilates methods before testing whether Pilates itself improves performance regardless of the modality.

Material and methods

- The description of the sample indicates that the players are amateurs but does not indicate in which league they play, if they all belong to the same team, how much training load they have, and all the variables required in this type of study. To improve this point, I suggest that they be guided by published works in this respect in which all the characteristics of the participants and their conditions are indicated in detail.

- The sample is small and by convenience, which presents a very important limitation in the results.

- The authors state that the participants have no injuries or health problems, but do not specify how this data was obtained.

- It is also specified that the players train regularly, but the workload is not specified.

- This section is disorderly and chaotic, which makes it difficult to read.

- The exercise protocol specifies that the control group does not do any exercise while the other two groups do a protocol. This biases the results as the control group members receive less training than the groups receiving Pilates. In other words, there are two groups that do 6 sessions per week while the control group does only 3. This already generates differences, regardless of the techniques used.

- The authors state that the distribution of the data is normal, but do not specify the test used or add the test coefficients.

- It is not clearly specified how the analyses have been carried out, as there is a pre-post and a comparison between groups. If all this has been done at the same time, it is more advisable to use an ANCOVA.

- With the sample size, I consider it much more accurate to perform non-parametric tests such as the Kruskal Wallis analysis, as ANOVA tests require a larger number of subjects. In order to use a two-way ANOVA, the groups must consist of at least 15 subjects, otherwise the results may be compromised.

Results

- The results are confusing and disorganised

I hope that these indications will help you to improve your work.

Kind regards

Reviewer #2: Overview:

The article presents a randomized controlled trial comparing the effects of reformer pilates (RP) and mat pilates (MP) exercises on the physical performance and technical skills of soccer players. The study reveals that both pilates methods improve the performance of soccer players, but reformer pilates is more effective than mat pilates. The article provides valuable information, especially for those working in the field of sports science and training methodology.

Thank you for the interesting insights your article provides. The manuscript is well-written and deals with an important topic. I have thoroughly reviewed the manuscript and would like to make some suggestions for your review.

Abstract section:

- RP group (n=10; 20.60 ± 1.65), MP group (n=10; 19.40 ± 1.35), and control group (CG) (n=10; 20.10 ± 1.15).

- Add unit, example: (age 20.60 ± 1.65 years)

Introduction section:

- “Soccer players must have a high degree of athletic performance characteristics such as including balance, coordination, flexibility, agility, speed, endurance, and technical

skills in order to properly accomplish these activities. Pilates exercises are used as a training method to improve these features”

add a reference to this sentence.

- add a hypothesis sentence

method:

Learn more about the study design. When and in what order were the tests performed? Indicate what time of the day the training and tests were performed.

Learn more about soccer training

Learn more about the “German Agility” test protocol.

Reviewer #3: Thank you for the opportunity to review the manuscript entitled - Randomised Controlled Study on the Effects of Pilates Exercises in Soccer: Comparing Mat and Reformer Methods on Physical and Technical Performance

The study is interesting and addresses a topical scientific topic.

Recommendations for improving the content of the manuscript:

Introduction:

• To detail the connection between pilates and soccer with a focus on the physical and technical components of the study. To argue why pilates was chosen and not other aerobic gymnastics or stretching programs.

• To detail more specifically the novel aspects of the present study in relation to previous studies on the same topic.

Materials and Methods:

• Participant and Randomisation – this section should be restructured, as it contains repeated details.

• Study design – to add this subsection where the periodization of the study will be detailed: year, testing periods, and stages of the study (part of the content of this section is also found in the Introduction and Randomization)

Result – this section is well structured and interpreted.

Discussions:

- To add at the end of the Discussions - the practical implications and limits of the study based on the relevant results.

Conclusions

- They recommended separating the text from the Discussions and including the Conclusions section, as well as the separate section at the end of the study.

6. PLOS authors have the option to publish the peer review history of their article (what does this mean? ). If published, this will include your full peer review and any attached files.

**Do you want your identity to be public for this peer review?** For information about this choice, including consent withdrawal, please see our Privacy Policy .

Reviewer #1: No

Reviewer #2: No

Reviewer #3: No

---

## [Author Response · Author response to Decision Letter 1]

28 Mar 2025

Dear referees, we would like to express our appreciation for your contributions to the article. The revisions we have mentioned have been made with precision by the authors. The answers are available in the Response to Reviewers file. The revisions are also included in the revised version of the article.

---

## [Decision Letter · Decision Letter 1]

22 Apr 2025

Randomised Controlled Study on the Effects of Pilates Exercises in Soccer: Comparing Mat and Reformer Methods on Physical and Technical Performance

PONE-D-25-04295R1

Dear Dr. Baťalík,

We’re pleased to inform you that your manuscript has been judged scientifically suitable for publication and will be formally accepted for publication once it meets all outstanding technical requirements.

Kind regards,

Julio Alejandro Henriques Castro da Costa

Academic Editor

PLOS ONE

Additional Editor Comments (optional):

Reviewers' comments:

Reviewer's Responses to Questions

**Comments to the Author**

1. If the authors have adequately addressed your comments raised in a previous round of review and you feel that this manuscript is now acceptable for publication, you may indicate that here to bypass the “Comments to the Author” section, enter your conflict of interest statement in the “Confidential to Editor” section, and submit your "Accept" recommendation.

Reviewer #2: All comments have been addressed

Reviewer #3: All comments have been addressed

2. Is the manuscript technically sound, and do the data support the conclusions?

Reviewer #2: Yes

Reviewer #3: Yes

3. Has the statistical analysis been performed appropriately and rigorously? 

Reviewer #2: Yes

Reviewer #3: Yes

4. Have the authors made all data underlying the findings in their manuscript fully available?

Reviewer #2: No

Reviewer #3: Yes

5. Is the manuscript presented in an intelligible fashion and written in standard English?

Reviewer #2: Yes

Reviewer #3: Yes

6. Review Comments to the Author

Reviewer #2: The article has undergone a comprehensive revision process, wherein all necessary corrections—including grammatical, stylistic, and content-related adjustments—have been meticulously addressed. Each modification has been applied in a manner consistent with scholarly standards, resulting in an improved and polished final version

Reviewer #3: The authors improved the manuscript with the title - ” Randomised Controlled Study on the Effects of Pilates Exercises in Soccer: Comparing Mat and Reformer Methods on Physical and Technical Performance” with the recommandations.

7. PLOS authors have the option to publish the peer review history of their article (what does this mean? ). If published, this will include your full peer review and any attached files.

**Do you want your identity to be public for this peer review?** For information about this choice, including consent withdrawal, please see our Privacy Policy .

Reviewer #2: No

Reviewer #3: No

---

## [Editor Report · Acceptance letter]

PONE-D-25-04295R1

PLOS ONE

Dear Dr. Baťalík,

I'm pleased to inform you that your manuscript has been deemed suitable for publication in PLOS ONE. Congratulations! Your manuscript is now being handed over to our production team.

Kind regards,

on behalf of

Dr. Julio Alejandro Henriques Castro da Costa

Academic Editor

PLOS ONE